# Dental Anomalies in Maxillary Incisors and Canines among Patients with Total Cleft Lip and Palate

Anna Paradowska-Stolarz * and Beata Kawala

Division of Dentofacial Anomalies, Department of Orthodontics and Dentofacial Orthopedics,
Wroclaw Medical University, Krakowska 26, 52-425 Wrocław, Poland; beata.kawala@umw.edu.pl
* Correspondence: anna.paradowska-stolarz@umw.edu.pl

**Abstract:** Cleft lip and palate is the most common asymmetric congenital condition of the orofacial region, which also finds its reflection in dental anomalies. The aim of the study was to present the dental asymmetries of the front region of the maxilla in cleft patients. **Materials and Methods:** We analyzed plaster casts and panoramic X-rays of 154 patients with total clefts and 151 healthy individuals. The cleft patients' age ranged between 7.1 and 20 years (mean 13.18). The control group had a similar age range (7.1 and 20 years, mean 13.44). The digital caliper was used to measure the width of the teeth. Each measurement was performed three times each. **Results:** Most of the dental anomalies among cleft patients referred to the lateral incisors and were focused on the cleft side. The asymmetry of the incisors is reflected both in the number of teeth in the cleft region and their width. The lateral incisor was missing twice as frequently on the cleft side of the individual. If present, the lateral incisor was usually ±1.5 mm narrower than the incisor on the opposite side. In bilateral clefts, dental anomalies occurred more frequently on the left side. **Conclusions:** Dental problems occurred more frequently in patients with total cleft lip and palate than in healthy individuals. The most commonly affected teeth were the lateral incisors. The width of the lateral incisors was reduced in cleft patients—showing a smaller mesiodistal dimension on the cleft side.

**Keywords:** cleft lip and palate; total cleft; asymmetry; symmetry; incisors; canines; dental esthetics

## 1. Introduction

Cleft lip and/or palate is the most common congenital deformity, with unilateral cleft lip and palate (UCLP) being the commonly recognized asymmetrical form. However, it is important to note that bilateral cleft lip and palate (BCLP) can also exhibit asymmetry, often due to the rotation of the incisive bone [1]. While genetics plays a critical role in the etiology of cleft lip and/or palate, the exact cause is multifactorial [2]. Additionally, differences in the oral microbiome between patients with clefts and those without this congenital deformity have been observed. These differences can lead to microbial imbalances, which in turn can result in challenges related to healing, increased risk of caries, and other conditions that impact overall health [1,3].

Patients with cleft lip and/or palate require multidisciplinary treatment involving specialists from various disciplines [1]. Most of the treatment focuses on the lip and nose, which are typically the most affected areas [4,5]. Surgical intervention forms the cornerstone of treatment, while cleft palate care focuses on collaboration among a range of specialists, including pediatricians, plastic surgeons, orthognathic surgeons, dentists, speech therapists, and orthodontists. The success of treatment relies heavily on building trust and fostering cooperation with the patient's family, especially the parents [1,6].

Among the challenges faced by patients with cleft lip and/or palate, severe malocclusions are a common issue. The predominant malocclusion observed in cleft patients is a crossbite, which has been reported to affect 75% of individuals with BCLP and 65% of those with UCLP, according to recent research [7]. Another frequently encountered malocclusion

is class III, primarily caused by maxillary hypoplasia. Many of these malocclusions may eventually necessitate orthognathic surgery due to underlying maxillary hypoplasia [8]. In an attempt to minimize the need for future orthognathic surgery, trials have been conducted using maxillary traction distractors [9]. It is important to consider whether this phenomenon is caused by maxillary hypoplasia or asymmetric mandibular growth, as both have been identified as genetically based causes [10]. Orthodontic treatment is typically required for several years, usually starting shortly after birth and lasting until adulthood. In the deciduous and mixed dentition stages, removable appliances are primarily employed, whereas full orthodontic treatment involving fixed appliances is planned for permanent dentition [11].

In addition to malocclusions, dental anomalies are another significant issue among cleft patients. These anomalies encompass not only the positioning of teeth, but also their shape, quality, quantity, and eruption timing. Dental anomalies, such as enamel hypoplasia and supernumerary teeth, are commonly observed on the cleft side [12–14]. The presence of additional tooth buds or the absence of teeth altogether, as well as malformed tooth anatomy, pose a challenge in achieving arch symmetry. This may cause a challenge for experienced dentists to restore arch symmetry, achieve desirable dental outcomes, and create a natural smile. Furthermore, cleft patients have a higher predisposition to dental caries and inadequate oral hygiene, making them potential candidates for future esthetic dentistry and prosthetic treatment [12–16].

Teeth impaction, especially in the cleft area, is a common problem. The canine tooth is the most frequently impacted in cleft patients [17]. This condition can also be associated with dentigerous cysts, further complicating the treatment [18]. Patients easily notice the asymmetry of their teeth, emphasizing the importance of addressing and restoring symmetry as a key aspect of orthodontic, prosthetic, and restorative treatment [19]. Successful prosthetic restoration requires not only perfect esthetics, but also proper function, often necessitating the use of facebows and articulators [20]. However, achieving esthetic and functional restoration in cleft patients can be challenging due to various types of malocclusion and a tendency for relapse after orthodontic treatment. The presence of soft tissues, such as Simonart's band, also adds complexity to the treatment process [7,21,22].

Many cleft patients require surgical-prosthodontic treatment in adulthood. In cases where tooth buds are missing, dental implant placement is often necessary, presenting a challenging procedure, especially when a severe bone deficiency exists in the cleft area [1,23]. A technique worth considering for assessing bone loss levels before implantation planning is fractal dimension analysis and bone index assessment [24]. Proper prosthetic treatment should also consider not only tooth color, but also gingival esthetics. Although the gingival margin is typically thinner and bone levels lower in cleft patients, an experienced surgeon and prosthodontist can work towards achieving satisfactory esthetic outcomes [25].

The purpose of this study was to present the problem of dental asymmetries observed in patients with cleft lip and palate. In the present study, we decided to focus on the maxillary front region since this is the most challenging part of regaining smile esthetics and achieving symmetry for individuals. The front region of the maxilla is easily accessible, and the expectations of the restorative outcome are the highest. The rehabilitation of the oral cavity in cleft patients is a challenging procedure, including surgeries, orthodontic preparations, and as a final procedure—prosthodontic treatment. Considering the occlusal problems commonly encountered in cleft patients, this study aimed to reveal the problem of dental asymmetries and highlight potential teeth restorations that would be part of the treatment planning process involving dentists. Based on our experience, most of these patients require dental restorations once the orthodontic procedures have been completed. For this paper, we decided to focus on the frontal region of the maxilla since it is the area most affected by dental anomalies. We focused on the presence or lack of teeth and their mesiodistal width to predict the feasibility of future esthetic restorations and anticipate the potential problems that arise from asymmetries in tooth quality and quantity.

Therefore, this study primarily aimed to present dental anomalies in the front maxillary region, encompassing both qualitative aspects, such as deformations (e.g., microdontia, peg-shaped teeth), and quantitative aspects, such as hypodontia and hyperdontia. Additionally, we sought to highlight the differences in tooth widths as a distinct marker of asymmetry.

## 2. Materials and Methods

### 2.1. Design and Settings

The study examined the medical records of a total of 2372 patients with clefts and 534 healthy individuals. These records were obtained from three medical centers in Poland: Wroclaw Medical University, Poznan Medical University, and Polanica Zdroj Hospital. The research was carried out by two investigators, with BK serving as the supervisor. The study duration spanned 4 years. Individuals with coexisting malformations and congenital syndromes were excluded from the research, as these conditions could potentially contribute to additional dental malformations. Only complete medical records, including panoramic X-rays for assessing tooth quantity (hypodontia, hyperdontia) and identification of impacted teeth, were considered.

The study focused exclusively on patients with isolated total clefts of the lip, alveolar bone, and palate. Only individuals who had both panoramic X-rays and dental casts from a similar time frame (with a maximum difference of 3 months between records) were included. Those without panoramic X-rays and/or dental casts were excluded from the research. This qualifying selection was implemented to prevent potential errors where the absence of a tooth bud on a dental cast might not necessarily indicate hypodontia. By eliminating such errors, the study aimed to provide a more accurate representation of the actual prevalence of dental anomalies among the examined patients.

Among the investigated medical records, cases of left-sided (L-CLP), right-sided (R-CLP), and bilateral (BCLP) clefts were included. The research protocol was approved by the Bioethical Committee of Wroclaw Medical University, Poland (KB—597/2008).

The research was conducted on patients aged between 7.1 and 20 years, with a mean age of 13.18. This age range was chosen because we specifically selected patients who had not undergone previous treatment with fixed orthodontic appliances and were enrolled in the orthodontic care program for craniofacial deformities. The control group was similarly aged, ranging from 7 to 20 years with a mean age of 13.44. The purpose of selecting a similar age range for the control group was to ensure comparability with the cleft patient group and include random individuals requiring orthodontic treatment.

Based on the selection criteria, the study included a total of 154 patients with clefts and 151 individuals in the control group. Table 1 presents the breakdown of the examined groups. The cleft patients were further divided into three separate subgroups, comprising individuals with bilateral cleft lip and palate (BCLP), left-sided cleft lip and palate (L-CLP), and right-sided cleft lip and palate (R-CLP). The fourth one consisted of individuals in the control group.

**Table 1.** The structure of the examined groups (BCLP—bilateral cleft lip and palate; L-CLP—unilateral, left-sided cleft lip and palate; and R-CLP—unilateral, right-sided cleft lip and palate).

|  | BCLP | L-CLP | R-CLP | Control |
|---|---|---|---|---|
| Females | 17 (11.1%) | 36 (23.4%) | 8 (5.2%) | 96 (64%) |
| Males | 19 (12.3%) | 51 (33.1%) | 23 (14.9%) | 55 (36%) |
| Sum | 36 (23.4%) | 87 (56.5%) | 31 (20.1%) | 151 (100%) |

The researchers formulated several hypotheses for the study:

- The asymmetry of the incisors would be evident in their quantity measurements.
- The asymmetry of the incisors would be observed in the mesiodistal measurement of tooth width.

- In unilateral clefts, the cleft side would exhibit a higher prevalence of dental anomalies compared to the opposite side.
- In bilateral clefts, dental anomalies would be equally present on both sides.

### 2.2. Methods Description

Both panoramic X-rays and plaster models were utilized in the study to evaluate the dental records of the individuals. These records provided information regarding the lack of teeth buds and any additional teeth. They also facilitated the assessment of the anatomical symmetry or asymmetry of the three front teeth, namely the central and lateral incisors, as well as the canines.

Since the researchers examined medical records, all measurements were conducted indirectly on plaster casts of the individuals. The mesiodistal width of the teeth was measured at the widest point of each tooth using a digital caliper. To minimize potential errors, each measurement was performed three times, and the arithmetic mean of these measurements was calculated. The resulting value was rounded to the second decimal place. The panoramic X-ray images were utilized to determine the number of tooth buds, providing a quantitative assessment.

These measurements allowed for the assessment of various dental malformations, such as tooth impaction, microdontia, and teeth malformations, as well as the evaluation of tooth widening or narrowing. The paper adheres to the requirements of STROBE guidelines, ensuring appropriate planning and construction [26]. The quality measurements were performed using a caliper, and the measurement methods are illustrated in Figure 1 to emphasize the investigative approach.

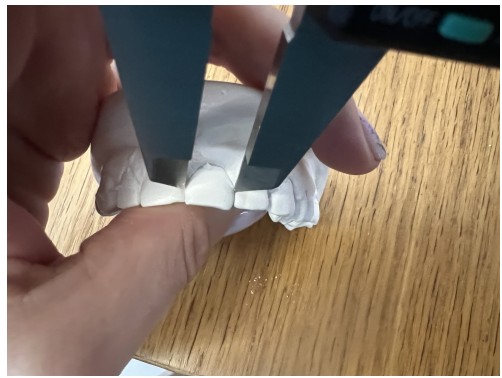 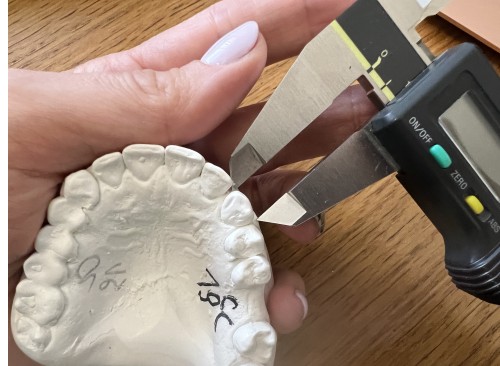

**Figure 1.** Presentation of the methodology of measurements of mesiodistal width of the teeth on the plaster casts. On the **left**—measurements from the front, on the **right**—from the occlusal side.

### 2.3. Statistical Analysis

The statistical evaluation of the research was conducted using Statistica v.10.0 (Tibco). The presence of both quality and quantity dental anomalies was established by calculating the arithmetic mean of the obtained data. Mean values of the front teeth (canines and incisors) were determined, followed by a correlation analysis and Student's *t*-test (with a significance level of $p < 0.05$). To assess statistically significant differences between the examined groups, ANOVA tests (variance analysis) were performed. Post hoc NIR tests were conducted only for statistically significant values. The threshold for statistical significance was set at $p < 0.05$. Separate *p*-values were provided for each sex. These *p*-values were used to compare all the examined groups with each other and are presented as results in the Supplementary Files. All values were rounded to the second decimal place.

## 3. Results

A total of 154 individuals with clefts and 151 individuals without facial deformities were included in the study. The mean age of the cleft patients was 13.18 years, while the mean age of the healthy individuals was 13.44 years.

Tables 2–5 present the occurrence of numerical (hypodontia and hyperdontia) and anatomical (malformed teeth, microdontia, and impacted teeth) dental anomalies in the four examined groups. The percentages were counted from the actual number of the present teeth—if one was missing, the total number of the examined teeth was decreased by one. Tables 2 and 3 refer to the male group, while Tables 4 and 5 refer to the female group.

In male patients with a unilateral cleft, the most common deformities were observed in the lateral incisor of the cleft side, which contributed to the majority of the observed asymmetries. Hypodontia was the most prevalent issue, with the lateral incisor missing in 35.29% of individuals in the L-CLP group and 43.5% in the R-CLP group. Anatomical malformation affected over 90% of the present lateral incisors in both groups. Similar to unilateral clefts, patients with bilateral clefts also exhibited quantity and quality deformities primarily in the lateral incisor region of the maxilla. Interestingly, left-sided hypodontia was observed nearly twice as often as right-sided hypodontia. Additionally, in the BCLP group, hyperdontia was more frequent in the lateral incisor region.

**Table 2.** Presence of quantity and quality dental anomalies in male patients with CLP (left-sided L-CLP, right-sided R-CLP, and bilateral BCLP).

| Tooth | Hypodontia N (%) | Hyperdontia N (%) | R-CLP (*n* = 23) Microdontia N (% of Present) | Deformed Tooth N (% of Present) | Impacted Tooth N (% of Present) |
|---|---|---|---|---|---|
| 13 | 0 | 0 | 0 | 0 | 0 |
| 12 | 10 (43.48) | 3 (13.04) | 9 (69.23) | 3 (23.08) | 2 (15.38) |
| 11 | 1 (4.35) | 0 | 2 (9.10) | 1 (4.55) | 1 (4.55) |
| 21 | 0 | 0 | 0 | 0 | 0 |
| 22 | 5 (21.74) | 0 | 1 (5.56) | 0 | 0 |
| 23 | 0 | 0 | 0 | 0 | 0 |
| | | | L-CLP (*n* = 51) | | |
| 13 | 0 | 0 | 0 | 0 | 0 |
| 12 | 7 (13.73) | 1 (1.96) | 1 (2.27) | 1 (2.27) | 0 |
| 11 | 0 | 0 | 0 | 0 | 1 (1.96) |
| 21 | 1 (1.96) | 0 | 2 (4) | 0 | 0 |
| 22 | 18 (35.29) | 9 (17.65) | 23 (69.7) | 10 (30.3) | 4 (12.12) |
| 23 | 0 | 0 | 0 | 0 | 9 (17.65) |
| | | | BCLP (*n* = 19) | | |
| 13 | 0 | 0 | 0 | 0 | 1 (5.26) |
| 12 | 5 (26.32) | 5 (26.32) | 8 (53.33) | 3 (20) | 2 (13.33) |
| 11 | 1 (5.26) | 0 | 0 | 0 | 0 |
| 21 | 0 | 0 | 0 | 0 | 0 |
| 22 | 9 (47.37) | 4 (21.05) | 4 (44.44) | 0 | 0 |
| 23 | 0 | 0 | 0 | 0 | 1 (5.26) |

**Table 3.** Presence of quantity and quality dental anomalies in male patients without cleft (*n* = 55).

| Tooth | Hypodontia N (%) | Hyperdontia N (%) | Microdontia N (% of Present) | Deformed Tooth N (% of Present) | Impacted Tooth N (% of Present) |
|---|---|---|---|---|---|
| 13 | 0 | 0 | 0 | 0 | 0 |
| 12 | 1 (1.82) | 0 | 0 | 0 | 0 |
| 11 | 0 | 0 | 0 | 0 | 0 |
| 21 | 0 | 0 | 0 | 0 | 0 |
| 22 | 1 (1.82) | 0 | 0 | 0 | 0 |
| 23 | 0 | 0 | 0 | 0 | 0 |

**Table 4.** Presence of quantity and quality dental anomalies in female patients with CLP (left-sided L-CLP, right-sided R-CLP, and bilateral BCLP).

| Tooth | Hypodontia N (%) | Hyperdontia N (%) | Microdontia N (% of Present) | Deformed Tooth N (% of Present) | Impacted Tooth N (% of Present) |
|---|---|---|---|---|---|
| **R-CLP (n = 8)** | | | | | |
| 13 | 0 | 0 | 0 | 0 | 0 |
| 12 | 4 (50) | 1 (12.5) | 3 (75) | 1 (25) | 0 |
| 11 | 0 | 0 | 1 (12.5) | 0 | 0 |
| 21 | 0 | 0 | 0 | 0 | 0 |
| 22 | 2 (25) | 1 (12.5) | 2 (33) | 1 (16.67) | 0 |
| 23 | 0 | 0 | 0 | 0 | 0 |
| **L-CLP (n = 36)** | | | | | |
| 13 | 0 | 0 | 0 | 0 | 1 (2.78) |
| 12 | 9 (25) | 0 | 0 | 0 | 0 |
| 11 | 0 | 0 | 0 | 0 | 0 |
| 21 | 1 (2.78) | 1 (2.78) | 0 | 0 | 0 |
| 22 | 17 (47.22) | 4 (11.11) | 15 (83.33) | 1 (5.56) | 3 (16.67) |
| 23 | 0 | 0 | 0 | 0 | 0 |
| **BCLP (n = 17)** | | | | | |
| 13 | 0 | 0 | 0 | 0 | 1 (5.88) |
| 12 | 7 (41.18) | 3 (17.65) | 7 (70) | 1 (10) | 0 |
| 11 | 0 | 1 (5.88) | 0 | 0 | 0 |
| 21 | 0 | 0 | 0 | 0 | 0 |
| 22 | 5 (29.41) | 5 (29.41) | 8 (66.67) | 2 (16.67) | 1 (8.33) |
| 23 | 0 | 0 | 0 | 0 | 0 |

**Table 5.** Presence of quantity and quality dental anomalies in female patients without cleft (*n* = 99).

| Tooth | Hypodontia N (%) | Hyperdontia N (%) | Microdontia N (% of Present) | Deformed Tooth N (% of Present) | Impacted Tooth N (% of Present) |
|---|---|---|---|---|---|
| 13 | 0 | 0 | 0 | 0 | 2 (2.08) |
| 12 | 5 (5.21) | 0 | 0 | 0 | 0 |
| 11 | 0 | 0 | 0 | 0 | 0 |
| 21 | 0 | 0 | 0 | 0 | 0 |
| 22 | 5 (5.21) | 0 | 0 | 0 | 0 |
| 23 | 0 | 0 | 0 | 1 (1.04) | 0 |

In contrast to the cleft group, the individuals without the congenital disease exhibited only one dental anomaly, which was hyperdontia of the lateral incisor. This finding was presented in Table 3. Among the male patients without clefts, none of them experienced canine impaction or deformations of the anatomical features of any teeth. The results indicate that asymmetries in quality and quantity are observed much more frequently among male patients with clefts compared to those without congenital anomalies.

Similar to the male group, among females, dental anomalies primarily affected the lateral incisor of the clefted side. Hypodontia of this tooth was twice as frequent as on the contralateral side of the dental arch. When discussing the occurrence of quantity and quality dental anomalies in females with BCLP, it was found that, unlike other groups, the most common maxillary dental problem was microdontia of the lateral incisor. Hyperdontia of the lateral incisor was more frequently observed on the left side, while hypodontia was more prevalent on the right side. Generally, the observations of asymmetries between the sexes were similar, with the majority of numerical malformations occurring on the cleft side. Interestingly, hypodontia of the lateral incisor was twice as common in healthy females compared to males.

Overall, the situation in the group of males and females without clefts was similar. Dental anomalies in the frontal region of the maxilla were less frequently observed compared to the group of patients with clefts. Among females without clefts, no additional or microdontic teeth were observed, and hypodontia affected only around 5% of the lateral incisors in this group. Canine impaction was more frequent on the right side in two females.

In both females and males, dental anomalies were found to occur more frequently in individuals with clefts compared to those without the condition. Furthermore, there was a predominance of dental anomalies on the clefted side. The lateral incisor was identified as the most affected tooth in these cases.

Table 6 presents the values of the widths of the maxillary incisors and canines. In patients with R-CLP, it was observed that the lateral right incisor was the narrowest tooth in the dental arch. The mean width of tooth 12 was more than 1 mm lower than that of tooth 22, indicating a significant level of asymmetry. In contrast, among L-CLP patients, tooth 22 had a mean width that was 1.5 mm lower than that of tooth 12, indicating an even higher level of asymmetry. In both cases, the asymmetry in tooth width affected the cleft side. Among males with BCLP, the left lateral incisor was narrower than the right lateral incisor by an average of 0.75 mm.

**Table 6.** The width of the teeth in the group of males.

| Tooth | Number of Samples | Mean | Minimum Value | Maximum Value | Standard Deviation |
|---|---|---|---|---|---|
| **No Deformity** | | | | | |
| 13 | 148 | 7.88 | 6.55 | 9.33 | 0.55 |
| 12 | 122 | 6.62 | 3.60 | 8.00 | 0.95 |
| 11 | 146 | 8.71 | 6.00 | 10.15 | 0.66 |
| 21 | 147 | 8.64 | 6.35 | 10.14 | 0.66 |
| 22 | 110 | 6.40 | 2.98 | 8.24 | 0.94 |
| 23 | 140 | 7.86 | 6.04 | 9.16 | 0.59 |
| **R-CLP** | | | | | |
| 13 | 23 | 7.64 | 6.55 | 8.76 | 0.59 |
| 12 | 11 | 5.63 | 3.82 | 6.82 | 0.93 |
| 11 | 21 | 8.37 | 6.93 | 9.52 | 0.69 |
| 21 | 23 | 8.54 | 7.31 | 10.02 | 0.73 |
| 22 | 18 | 6.81 | 5.50 | 8.11 | 0.75 |
| 23 | 23 | 7.68 | 6.61 | 8.87 | 0.59 |
| **L-CLP** | | | | | |
| 13 | 51 | 7.93 | 7.06 | 9.33 | 0.50834 |
| 12 | 44 | 6.98 | 4.49 | 8.00 | 0.65669 |
| 11 | 51 | 8.73 | 7.73 | 10.15 | 0.56296 |
| 21 | 50 | 8.42 | 6.50 | 9.46 | 0.57413 |
| 22 | 30 | 5.56 | 2.98 | 8.24 | 1.01947 |
| 23 | 45 | 7.92 | 6.23 | 9.11 | 0.53647 |
| **BCLP** | | | | | |
| 13 | 23 | 7.64 | 6.55 | 8.76 | 0.59 |
| 12 | 11 | 5.63 | 3.82 | 6.82 | 0.93 |
| 11 | 21 | 8.37 | 6.93 | 9.52 | 0.69 |
| 21 | 23 | 8.54 | 7.31 | 10.02 | 0.73 |
| 22 | 18 | 6.81 | 5.50 | 8.11 | 0.75 |
| 23 | 23 | 7.68 | 6.61 | 8.87 | 0.59 |

To provide a broader understanding of the results, a variance analysis of teeth width was conducted in the male groups, and all the presented results were found to be statistically significant, as shown in Table 7.

Following the variance analysis, six Supplementary Figures (Figures S1–S6) were prepared to visually illustrate the comparison of widths between canines and incisors in the examined male groups. The statistical analysis revealed that the differences in tooth widths between healthy individuals and cleft patients were not only related to symmetry, but also varied among the different groups. Generally, the most significant differences were observed among the incisors.

**Table 7.** Variance analysis in tooth dimensions in the group of males (SS—sum of squares, df—degrees of freedom, MS—mean sum of squares, F—test for variances differences, *p*—statistical significance). All the statistically significant values are presented in red.

| Tooth | SS—Effect | df—Effect | MS—Effect | SS—Error | df—Error | MS—Error | F | p |
|---|---|---|---|---|---|---|---|---|
| **13** | 6.071 | 3 | 2.024 | 37.8 | 144 | 0.263 | 7.70223 | 0.000083 |
| **12** | 48.673 | 3 | 16.224 | 59.8 | 118 | 0.506 | 32.03587 | 0.000000 |
| **11** | 9.043 | 3 | 3.014 | 53.7 | 142 | 0.378 | 7.97142 | 0.000060 |
| **21** | 7.951 | 3 | 2.650 | 55.7 | 143 | 0.390 | 6.79829 | 0.000257 |
| **22** | 34.837 | 3 | 11.612 | 60.8 | 106 | 0.574 | 20.24232 | 0.000000 |
| **23** | 7.071 | 3 | 2.357 | 41.9 | 136 | 0.308 | 7.65669 | 0.000091 |

The differences in the widths of the teeth 13 were observed in various comparisons among male patients with unilateral clefts (Figure S1). Similar differences were observed when comparing patients with L-CLP to the group with BCLP. Statistically significant differences were noted when compared to the R-CLP and BCLP groups, with the control group serving as a reference.

Figure S2 demonstrates the differences in the width of the lateral right incisor. Statistically significant differences were observed between males with R-CLP and L-CLP, as well as between L-CLP and BCLP. However, when compared to the control group, statistically significant differences were not observed in the L-CLP group.

Figure S3 presents statistically significant differences in the widths of tooth 11 among patients with L-CLP and R-CLP, as well as between L-CLP and BCLP. Again, when compared to the control group, males with L-CLP did not exhibit statistically significant differences.

Similar results were observed in Figure S4, where the other central incisor was compared.

In Figure S5, statistically significant differences were observed in the widths of tooth 22 between L-CLP and R-CLP patients, as well as between R-CLP and BCLP. Additionally, significant differences were found when comparing L-CLP and BCLP patients to the control group.

Finally, Figure S6 presents statistically significant differences in the width of tooth 23. Correlations were observed among patients with L-CLP when compared to R-CLP, and significant differences were also found between BCLP and R-CLP when compared to healthy individuals.

Table 8 provides an overview of the differences in tooth widths in the female group. Similar to the male group, the largest differences in width were observed in the lateral incisors. In unilateral cleft female patients, the most significant differences were found in the lateral incisor of the cleft side, with a difference of 1.5 mm compared to the unaffected side. In bilateral cleft patients, the widths of the teeth on both sides were comparable.

It is noted that the mean values of teeth widths are larger for incisors, while the widths of canines are comparable. Table 9 presents the results of the variance analysis of teeth width in the female group, demonstrating that all the presented results were statistically significant.

Following the variance analysis, Figures S7–S12 were prepared to visualize the statistically significant differences in teeth width within the female group. Figures S7 and S12 focus on differences among the canines, while Figures S8–S11 depict differences among the incisors. These figures are included in the Supplementary Materials.

Figure S7 illustrates the statistically significant differences in the widths of tooth 13, which were observed only when the group of L-CLP females was compared to the healthy individuals.

Figure S8 illustrates the correlation between the widths of tooth 12. Statistically significant values were observed between all groups of female patients with clefts when compared to healthy individuals. Among females without congenital deformities, the width of this tooth was statistically the highest.

**Table 8.** The width of the teeth in the group of females.

| Tooth | Number of Samples | Mean | Minimum Value | Maximum Value | Standard Deviation |
|---|---|---|---|---|---|
| | | **No Deformity** | | | |
| 13 | 154 | 7.60 | 6.31 | 9.40 | 0.52 |
| 12 | 132 | 6.59 | 4.42 | 8.28 | 0.70 |
| 11 | 157 | 8.45 | 6.55 | 9.95 | 0.62 |
| 21 | 155 | 8.41 | 6.87 | 9.87 | 0.60 |
| 22 | 123 | 6.35 | 2.95 | 8.30 | 0.91 |
| 23 | 156 | 7.59 | 6.16 | 9.43 | 0.57 |
| | | **R-CLP** | | | |
| 13 | 8 | 7.61 | 6.66 | 8.36 | 0.66 |
| 12 | 5 | 5.94 | 4.69 | 7.25 | 0.94 |
| 11 | 8 | 8.12 | 7.25 | 9.00 | 0.59 |
| 21 | 8 | 8.41 | 7.82 | 8.82 | 0.29 |
| 22 | 5 | 6.47 | 5.30 | 7.22 | 0.76 |
| 23 | 8 | 7.77 | 7.04 | 8.50 | 0.56 |
| | | **L-CLP** | | | |
| 13 | 35 | 7.27 | 6.31 | 8.39 | 0.53 |
| 12 | 26 | 6.59 | 5.68 | 7.40 | 0.45 |
| 11 | 36 | 8.21 | 7.12 | 9.56 | 0.64 |
| 21 | 34 | 8.07 | 6.87 | 9.32 | 0.61 |
| 22 | 16 | 5.07 | 3.61 | 6.50 | 0.86 |
| 23 | 35 | 7.31 | 6.16 | 9.43 | 0.66 |
| | | **BCLP** | | | |
| 13 | 16 | 7.52 | 6.74 | 8.00 | 0.38 |
| 12 | 10 | 5.50 | 4.42 | 6.95 | 0.74 |
| 11 | 17 | 8.20 | 6.55 | 8.87 | 0.62 |
| 21 | 17 | 8.19 | 7.21 | 9.09 | 0.61 |
| 22 | 11 | 5.60 | 2.95 | 7.81 | 1.30 |
| 23 | 17 | 7.70 | 6.81 | 8.46 | 0.49 |

**Table 9.** Variance analysis in tooth dimensions in the group of females (SS—sum of squares, df—degrees of freedom, MS—mean sum of squares, F—test for variances differences, *p*—statistical significance). All the statistically significant values are presented in red.

| Tooth | SS—Effect | df—Effect | MS—Effect | SS—Error | df—Error | MS—Error | F | p |
|---|---|---|---|---|---|---|---|---|
| **13** | 5.493 | 3 | 1.8309 | 36.2 | 150 | 0.241 | 7.59283 | 0.000092 |
| **12** | 16.003 | 3 | 5.3342 | 47.4 | 128 | 0.370 | 14.41055 | 0.000000 |
| **11** | 6.711 | 3 | 2.2370 | 53.6 | 153 | 0.351 | 6.38104 | 0.000420 |
| **21** | 6.873 | 3 | 2.2909 | 49.1 | 151 | 0.325 | 7.04449 | 0.000183 |
| **22** | 40.748 | 3 | 13.5825 | 59.5 | 119 | 0.500 | 27.14699 | 0.000000 |
| **23** | 3.801 | 3 | 1.2669 | 46.2 | 152 | 0.304 | 4.16365 | 0.007236 |

Figure S9 illustrates the statistically significant differences in the widths of tooth 11 among the groups of female patients with clefts when compared to healthy individuals. In healthy individuals, the width of this tooth was statistically the highest.

The statistically significant dependence between the widths of tooth 21 is presented in Figure S10. Correlations were observed between the groups of L-CLP and BCLP when compared to the patients without congenital deformities. In the patients without clefts, the width of tooth 21 was the highest.

Figure S11 shows the differences in the widths of tooth 22. Statistically significant values were observed between the left lateral incisor when the group of R-CLP was compared to the groups of L-CLP and BCLP. When compared to healthy individuals, differences were observed between the left-sided and bilateral clefts.

Finally, Figure S12 presents the differences in the widths of the left canines. A statistically significant difference was observed in the left maxillary canine when comparing the group of L-CLP patients to any other group in the research.

The presented data helped us to sustain the null hypotheses 1–3, but hypothesis 4 was rejected.

## 4. Discussion

The aim of this study was to present the issue of dental asymmetries in patients with cleft lip and palate. Due to the abundance of data available, we decided to focus on the maxillary front region. This is the challenging part of regaining the smile esthetics and sense of symmetry of an individual, making it a focal point for many dentists.

The researchers' initial observation centers around the prevalence of left-sided clefts among individuals with cleft lip and palate. This occurrence can be attributed to embryology, as the right side of the maxillary bone attaches to the premaxilla segment earlier, leading to a longer period required for closing the gap [27]. Consequently, if the closure process fails around the 8th week in utero, it offers an opportunity for the right side of the maxilla to join later in development.

Because of the complex etiology and the varied appearance of defects, patient care starts either during the pubertal period or shortly after birth. The treatment of cleft patients adopts a multidisciplinary approach aimed at regaining the function and symmetry of the individual [1].

Dental treatment initially starts with orthodontics, but it is crucial to plan the outcome in collaboration with a restorative or prosthetic dentist. Our research highlighted the considerable challenges associated with restoring perfect symmetry, especially since dental anomalies are most evident in the cleft region. Anatomical differences between the left and right sides of the dental arch might cause problems with the esthetic restoration and significantly influence the final result. Similar findings have been reported by other researchers as well [28,29].

Furthermore, studies have shown that these asymmetries extend beyond the incisors and canines, affecting the entire dentoalveolar arch. Many patients require prosthetic rehabilitation following surgical and orthodontic treatments, with the most symmetric arches observed when dental implants are utilized [30]. Additionally, Canadian researchers [31] have revealed that the most significant arch contraction occurs in the canine region of the clefted side. The presence of nasal and lip asymmetry, as demonstrated by Thierens et al. [32], further contributes to the perspective of dental asymmetry. It should be noted that our data, along with previously mentioned studies, were not categorized based on patient age, as the dental anomalies discussed do not vary over time.

Nasal asymmetry impacts facial perception and is observed in all patients with clefts in the lip area, thereby influencing the feasibility of prosthetic restoration for the upper arch. An intriguing alternative for restoring asymmetrically shaped teeth is the flow injection technique, also known as the injectable resin technique. This method helps minimize tooth preparation while addressing asymmetry [33].

The data obtained in our research will be helpful for general treatment planning and highlight the problems that may arise in achieving a perfect smile for patients with cleft deformities.

Asymmetry of the incisors is a commonly observed phenomenon that holds significant self-esteem implications for many individuals, including those with cleft lip and palate [19]. In terms of further investigation, establishing an asymmetry index based on 3D images of individuals could be a valuable approach [34,35]. However, for practical purposes, dental casts or 3D scans remain the most accessible methods for assessing teeth asymmetries [34]. Our research aligns with a previous study [36], which found that cleft individuals tend to have lower mesiodistal width of teeth.

Furthermore, similar findings from other studies indicate that most of the asymmetries in teeth structure are concentrated in the region of the lateral incisors [28,29]. Our paper also confirms this pattern, as the majority of dental anomalies were observed in that specific region, reflecting the highest degree of asymmetry. It is important to consider that the

maxilla tends to relapse after orthodontic widening, which should be considered during treatment planning [37].

Even orthognathic surgery, although considered the most reliable method, requires careful planning to restore not only the teeth, but also facial symmetry [38].

Dental asymmetries are a common issue among patients with clefts. Szyszka-Sommerfeld et al. [7] revealed that these asymmetries were also observed in the malocclusions, with crossbites on the affected side being the most frequent. Although this study did not specifically address overall occlusion, it revealed that dental asymmetries in terms of tooth size are present not only in the lateral incisors, but also in the central incisors and canines. This must be considered when planning for restorative or prosthetic treatment. To require that, the Bolton index, which measures the mesiodistal width of the front teeth in both the maxilla and mandible, could be utilized [39]. This study showed that the reduced mesiodistal width of teeth is reduced in the cleft patient group, and this might cause difficulties in treatment planning.

Our findings indicate that differences between sexes do not vary significantly when accounting for the type of cleft. However, there was a small problematic group consisting of females with right-sided clefts ($n = 8$), which aligns with similar observations reported in the literature [40]. Another study by Rubbo [41] reported a similar number of dental anomalies, although this study provides more detailed results by differentiating between the cleft and noncleft sides and categorizing anomalies based on cleft type. Considering these results, we believe that this approach provides greater accuracy. It was found that the right side is more frequently affected regardless of the type of cleft, which is particularly noticeable in cases of bilateral cleft. In contrast, Al Jamal et al. [42] observed a high prevalence of dental anomalies, but their study considered the entire dentition without specifically focusing on the front maxillary region. It is possible that these differences can be attributed to ethnic variations, as the study by Jamal et al. focused on the Jordanian population, while ours focused on the Polish population.

The research was conducted on many individuals from three independent centers. The research was performed by one person, which is one of the biggest advantages of this research. The individuals examined were not from a single center, indicating a high likelihood of encountering a variety of anomalies in our opinion.

A novel aspect of this research is the division into the left and right sides, which sheds light on the issue of dental asymmetry in cleft patients as a whole. However, it is important to acknowledge a limitation of the study, namely the lack of intraoral scans. Intraoral scans could potentially provide higher precision in measurements. However, due to the retrospective nature of the study and the large number of individuals involved, accessing dental scans for such a significant sample size was not feasible.

## 5. Conclusions

Patients with total cleft lip and palate are more prone to dental issues, particularly in the incisal and canine regions, compared to individuals without clefts. These dental anomalies affect both the quality and quantity of teeth, with a particular focus on the cleft area. Among the affected teeth, the lateral incisors are the most commonly impacted, often exhibiting hypodontia (missing teeth) and microdontia (smaller size). The width of lateral incisors is generally lower in cleft patients, and when comparing the unaffected side to the cleft side, the width is even lower in the latter. Given the higher incidence of dental malformations and overall asymmetry, conservative and prosthetic restoration of the smile can be more challenging for cleft patients compared to individuals without clefts.

**Supplementary Materials:** The following supporting information can be downloaded at: https://www.mdpi.com/article/10.3390/app13116635/s1, Figure S1: The comparison between the widths of teeth 13 between the boys (group 1—R-CLP, group 2—L-CLP, group 3—BCLP, group 4—healthy individuals. Statistically significant values were presented in red, Figure S2: The comparison between the widths of teeth 12 between the boys (group 1—R-CLP, group 2—L-CLP, group 3—BCLP, group 4—healthy individuals. Statistically significant values were presented in red, Figure S3: The comparison

between the widths of teeth 11 between the boys (group 1—R-CLP, group 2—L-CLP, group 3—BCLP, group 4—healthy individuals. Statistically significant values were presented in red, Figure S4: The comparison between the widths of teeth 21 between the boys (group 1—R-CLP, group 2—L-CLP, group 3—BCLP, group 4—healthy individuals. Statistically significant values were presented in red, Figure S5: The comparison between the widths of teeth 22 between the boys (group 1—R-CLP, group 2—L-CLP, group 3—BCLP, group 4—healthy individuals. Statistically significant values were presented in red, Figure S6: The comparison between the widths of teeth 23 between the boys (group 1—R-CLP, group 2—L-CLP, group 3—BCLP, group 4—healthy individuals. Statistically significant values were presented in red, Figure S7: The comparison between the widths of teeth 13 between the girls (group 1—R-CLP, group 2—L-CLP, group 3—BCLP, group 4—healthy individuals. Statistically significant values were presented in red, Figure S8: The comparison between the widths of teeth 12 between the girls (group 1—R-CLP, group 2—L-CLP, group 3—BCLP, group 4—healthy individuals. Statistically significant values were presented in red, Figure S9: The comparison between the widths of teeth 11 between the girls (group 1—R-CLP, group 2—L-CLP, group 3—BCLP, group 4—healthy individuals. Statistically significant values were presented in red, Figure S10: The comparison between the widths of teeth 21 between the girls (group 1—R-CLP, group 2—L-CLP, group 3—BCLP, group 4—healthy individuals. Statistically significant values were presented in red, Figure S11: The comparison between the widths of teeth 22 between the girls (group 1—R-CLP, group 2—L-CLP, group 3—BCLP, group 4—healthy individuals. Statistically significant values were presented in red, Figure S12: The comparison between the widths of teeth 23 between the girls (group 1—R-CLP, group 2—L-CLP, group 3—BCLP, group 4—healthy individuals. Statistically significant values were presented in red.

**Author Contributions:** Conceptualization, A.P.-S.; methodology, A.P.-S.; validation, A.P.-S. and B.K.; formal analysis, A.P.-S. and B.K.; investigation, A.P.-S.; resources, A.P.-S.; data curation, A.P.-S.; writing—original draft preparation, A.P.-S.; writing—review and editing, B.K.; visualization, A.P.-S.; supervision, A.P.-S. and B.K.; funding acquisition, A.P.-S. All authors have read and agreed to the published version of the manuscript.

**Funding:** This research received no external funding.

**Institutional Review Board Statement:** The research protocol was approved by the Bioethical Committee of Wroclaw Medical University, Poland (KB—597/2008).

**Informed Consent Statement:** Patient consent was waived, because we worked on the archival data (only the medical records were taken into account).

**Data Availability Statement:** Due to privacy we do not publish the detailed data of the patients, all are available at APS.

**Conflicts of Interest:** The authors declare no conflict of interest.

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
