# Peer review of "Dental Anomalies in Maxillary Incisors and Canines among Patients with Total Cleft Lip and Palate"

_applsci, doi:10.3390/app13116635_

Round 1

Reviewer 1 Report

Dear Author,

1. Please reduce the abstract to 200 word.

2. Most of the measurements included in the result are not mentioned in the discussion. Neither the male/female ratio nor the comparison of the results with international literature is described.

Without them, the discussion itself cannot be interpreted.

3. Please discuss previous findings in the international scientific literature regarding the association between patients without CLP and aplasia / microdontia. Compare this with your results.

Author Response

Dear Reviewer 1,

please find the answear to your suggestions in the attached file.

Reviewer 2 Report

The manuscript entitled "Considerations about anatomical symmetry of the incisors and canines in the maxilla in total cleft lip and palate patients" describes observational research on two groups, with and without cleft lip and palate. While it seems that the manuscript has been already modified, I recommend some additional modifications:

- The Materials and Methods section should be divided into several sections, such as (1) design and settings, (2) methods description, (3) statistical analysis. Also, I believe that on page 3, lines 127-131 should be included in the Results section, rather than in the Methods section. Some images on how the values were obtained on the models and Xrays would also be welcomed.

- The Results section should be also reorganized, so that the statistical analysis is easy to be followed and the tables included should also emphasize this process as well.

The discussion section should also emphasize better the clinical aspects of this research. In other words, how this article will help more exactly the practitioner who has to reconstruct the smile of a patient with cleft lip and palate.

Author Response

Dear Reviewer 2, 

thank you for an effort you gave into reviewing the paper. The changes were added in the additional file. Best regards

Reviewer 3 Report

Dear Authors,

I have completed my evaluation of the article. Although I appreciate your efforts, I found several important points that undermine the results of your study. Here are my comments:

General comments:
-The complete manuscript requires language correction.

For example:

1) It is recommended to write the sentences in the passive.

2) Please write "eg." as follows: e.g., microdontia...

3) Please correct "hipodontia" as follows: hypodontia

4) Indicated as “L-CLP and R-CLP” in the text but as “CLP-L- and CLP-R” in Table 2. Please correct these throughout the manuscript.

- Introduction:

1) This part should be revised based on the goals of the study.

2) After "the purpose of the study was…." This part should be included in the discussion.

3) Thus, please rewrite this part.

-Materials and methods:

1) Please revise as follows: The study was conducted by two investigators.

The study was conducted by one researcher (A.P-S.), and the second re-searcher (B.K.) was a supervisor.

- Results:

1) It is better to summarize results in three to four paragraphs. Then, the article will be beneficial to the dentists who read it. Thus, please provide short and useful results for all the readers.   

2) The text must not repeat the data from tables and/or figures.

3) If possible, please reduce the number of tables.

4) "When discussing the occurrence of quantity…" This part should be included in the discussion.

- Discussion: 

1) This part should begin with a brief interpretation of your study, followed by the findings of the study and comparisons with other studies. Please rewrite this part.

- Conclusion:

1) The conclusion must be brief and report the most important findings of the research. 

After the paper is revised, if possible, provide quantitative data among the significant results of your study in the results section of the abstract.

-The complete manuscript requires language correction.

For example:

1) It is recommended to write the sentences in the passive.

2) Please write "eg." as follows: e.g., microdontia...

3) Please correct "hipodontia" as follows: hypodontia

4) Indicated as “L-CLP and R-CLP” in the text but as “CLP-L- and CLP-R” in Table 2. Please correct these throughout the manuscript.

Author Response

Dear Reviewer 3, please find our responses in the attached file. Thank you for an effort in reviewing our paper. We hope the changes we applied would satisfy you. Thank you - Authors

Reviewer 4 Report

Orofacial clefts are common birth defects that cause life-long problems. Asymmetries and agenesis of the teeth that surround lateral cleft sites (i.e. lateral incisor and canine) frequently occur in individuals with cleft lip with or without cleft palate. Here, the authors retrospectively use casts and X-rays to analyze dentition patterns and tooth sizes in patients with CLP or normal controls, taking into account cleft sidedness, extent, and patient sex. The study is well designed and of interest to the scientific and medical communities, as well as affected patients and their families.

The work is sound and the analysis and interpretation are appropriate. However the manuscript would be improved by moderate to extensive English language editing, as several non-conventional uses of English grammar and terms are present in the manuscript. A few examples are Hipodontiaàhypodontia, weatheràwhether, and maksimum à maximum. I recommend seeking English editing for the manuscript. However, given the scientific merit of the work, my overall recommend is minor revision.

The quality of english language in the manuscript is moderate and would be benefited by english language editing.

Author Response

Dear Reviewer,

thank you for the positive feedback. The changes had been applied and the English had been corrected by the native speaker. Thank you for the support and revision of this manuscript - the time you spent on that is very important to us - Authors

Reviewer 5 Report

The present study focused on asymmetry of maxillary dental elements in palates of cleft palate patients and non cleft palate individuals.

Overall, although the good intentions of the authors easily understandable from the first reading, and the efforts spent in such large study sample, there is much confusion and lack of clearness on the aim of the study, the results obtained, and the critical aspects of the latter that are even poorly discussed and compared with the literature.

In the abstract there is not a clear purpose, which can surely be improved also in the introduction.

Since the authors refer more often to asymmetry instead of symmetry, and given the nature of the anomaly which is mostly asymmetrical per se, why don’t they refer to asymmetry also in the title?

The term ‘women’ and ‘men’ should be always changed to ‘females’ and ‘males’ (either in tables and in the text).

I am not sure on what has been evaluated in x-ray and what in the casts. Why the use of both (no justification).? 

Just looking at table 7, as example, I had much difficulties in understanding what the p refer to: are differences among the 4 groups (BCLP; RCLP; LCLP and control) or what? No clear explanation is given neither in the text nor in the table caption. Whay there is not the same table for the fameale sample? The table shoud be simplified and include also the F and p for the femal sample.

Many typos denoting a sloppy quality and the lack of an intensive revision by the authors which, I do not hide, made me surprised.

 Many sentences, especially the ones highlighted in yellow, require an extensive re-wording and in general all the manuscript should be revised by a native speaker. 

Results lacks of a clear focus and so the discussion does not really consider them critically, but lacks of contents, any comparisons with the literature and so of scientific critical view and soundness.

 Many sentences, especially the ones highlighted in yellow, require an extensive re-wording and in general all the manuscript should be revised by a native speaker. 

Author Response

Dear Reviewer, thank you for the suggestions. We hope that after applying them, the paper is more readable and understandable. The detailed reply was added in the additional file. Best regards - Authors

Reviewer 6 Report

This paper aims to evaluate incisal and canines’ anatomical symmetry in total cleft lip and palate patients. Maxillary teeth of cleft patients reveal more problems than control patients, being the lateral incisors the most affected teeth.

Either way, I may have some comments in the various sections:

The title should be simple and concise and describe the research. I think at the present is too vague and does not reflect a research paper.

The abstract – materials and methods should indicate the variables measured in each patient and respective scale of measurement. statistical tests and hypotheses should be described. Conclusions should be referred to the objectives of the study in the past tense ofverbs

Objectives- should be placed in the hypothesis that could be treated statistically (e.g. differences of width between cleft palates and control…or similar)

Material and Methods- All variables should be explained in detail. For example hypodontia was described as a qualitative variable (yes or no) or a quantitative one. Please describe the nature of each variable and the scale of measure.

Measurements of the mesiodistal width of teeth were performed on what surface? X-ray, please describe the method of measurement and type of variable and units with greater detail. Experiments should be described in a way that can easily be reproduced by any researcher.

Statistical analysis description should include homogeneity of variance (e.g. with Levene or Shapiro tests) of the distribution that was perform in order to submit data to parametric tests (ANOVA)

This paper aims to evaluate incisal and canines’ anatomical symmetry in total cleft lip and palate patients. Maxillary teeth of cleft patients reveal more problems than control patients, being the lateral incisors the most affected teeth.

Either way, I may have some comments in the various sections:

The title should be simple and concise and describe the research. I think at the present is too vague and does not reflect a research paper.

The abstract – materials and methods should indicate the variables measured in each patient and respective scale of measurement. statistical tests and hypotheses should be described. Conclusions should be referred to the objectives of the study in the past tense of verbs

Objectives- should be placed in the hypothesis that could be treated statistically (e.g. differences of width between cleft palates and control…or similar)

Material and Methods- All variables should be explained in detail. For example hypodontia was described as a qualitative variable (yes or no) or a quantitative one. Please describe the nature of each variable and the scale of measure.

Measurements of the mesiodistal width of teeth were performed on what surface? X-ray, please describe the method of measurement and type of variable and units with greater detail. Experiments should be described in a way that can easily be reproduced by any researcher.

Statistical analysis description should include homogeneity of variance (e.g. with Levene or Shapiro tests) of the distribution that was perform in order to submit data to parametric tests (ANOVA)

Results

Why are the results separated between males and females? There is some predominance described in the literature for this separation in results. Theis tables must be reviewed.

Author Response

Thank you for the effort in reviewing the paper. Our replies were added into the additional, attached file. We hope this changes would satisfy the Reviewer 6. 

Reviewer 7 Report

This paper is aimed to present the problem of dental asymmetries present in the cleft lip and palate patients from three medical centers in Poland.

Overall, this new version of the manuscript have addressed my previous concerns. Only minor language issues remain, to check in the editing stage.

Only minor language issues remain, to check in the editing stage. As the use of Maksimum in Tables.

Author Response

Dear Reviewer, thank you once again for the feedback and help in improving our paper. The paper this time was submitted to another journal, as we finally got the comment it did not fit the journals scope. Thank you for an effort in reviewing both versions of the manuscript. Further, the English quality had been revised by the native speaker. - Authors.

Reviewer 8 Report

The purpose of this study was to present the problem of dental asymmetries present in cleft lip and palate patients. The idea is interesting, but the implementation has flaws.

1. Modify the last paragraph of the introduction so that it does not mention "..the authors...."...but in this study....

2. Which ethics committee reviewed the study, one institution or all institutions?

3. How is the total size calculated?

4. In what period was the study conducted?

5. Present Table 1 with frequencies.

6. The idea of the work is not bad, but the presentation of the result is uninteresting. Descriptive anomalies by gender with or through CLP are listed.

It could be more interesting to compare the frequency of anomalies according to the demographic data of sex, age, CLP, and tooth width by means of a correlation.

7. Start the discussion with the purpose and hypothesis, whether it is confirmed or not, and only after that discuss the result by result.

8. What are the study's strengths and limitations?

9. Check 2, 3, 5, 6, 12, 13, 14, 16, 17, 18, etc. according to the instructions of the Journal.

Author Response

Dear Reviewer, 

thank you for time spent on the revision of our paper. Please, find the response to your suggestions in the attached file. Best regards - Authors

Round 2

Reviewer 1 Report

Dear Author,

Just one modification and is needed.

Please shorten the abstract, now it is 33% longer than 200 words.

It isn't just "only a bit" as you mentioned. 

The second thing, please mention the number of rows, where you discuss the difference "the association between patients without CLP and aplasia / microdontia" and the rows, where you are "Compare this with your results".

Best regards

Author Response

Dear Reviewer,

the abstract was reduced to 240 words now - we hope this satisfies the Reviewer. If not, I would need to ask the Editors to delete the introduction / background, as we are not able to present minimum of data of materials and methodology and results.

To be honest, I do not understand the second comment of the Reviewer. Thank you for the support - authors

Reviewer 2 Report

the modifications have been performed

Author Response

Thank you once again for the effort in reviewing the paper. - Authors

Reviewer 3 Report

Dear Authors,

I have completed my evaluation of the article.

These are my comments:

1) The manuscript is well revised.

2) Regarding dental anomalies in maxillary incisors and canines among patients with cleft lip and palate, the study provided satisfactory results.

3) Thus, the manuscript can be accepted for publication.

Author Response

Dear Reviewer, thank you once again for an effort in reviewing the manuscript. We think it really improved thanks to your suggestions. Best regards - authors

Reviewer 5 Report

The authors put some efforts to improved their manuscript whose revised version is now quite fine.

Author Response

(The authors gave the same response as above.)
